# Three Closed-Loop Feedback Control System with Dual Disturbance Observers of an Optoelectronic Stable Control Platform

**Hanwen Zhang** [1,2,3], **Yao Mao** [1,2,3], **Jiuqiang Deng** [1,2,3] and **Huabo Liu** [4,*]

[1] Key Laboratory of Optical Engineering, Chinese Academy of Sciences, Chengdu 610209, China; 15528446460@163.com (H.Z.); maoyao@ioe.ac.cn (Y.M.); jqdeng2016@126.com (J.D.)

[2] Institute of Optics and Electronics, Chinese Academy of Sciences, Chengdu 610209, China

[3] University of Chinese Academy of Sciences, Beijing 100049, China

[4] Department of Control Science and Engineering, School of Automation, Qingdao University, Qingdao 266071, China

[*] Correspondence: hbliu@qdu.edu.cn; Tel.: +86-532-8595-3672

**Abstract:** Disturbances presented in aeronautical imaging equipment can cause visual axis jitter, which directly leads to a reduction in closed-loop bandwidth and a decrease in tracking accuracy. The disturbance frequency affecting the stable control platform is mainly concentrated in the low- and middle-frequency bands, but the commonly used three closed-loop feedback control methods do not perform well in the disturbance rejection of those frequency bands. Moreover, the only disturbance observer in the acceleration loop cannot improve the low-band disturbance rejection capability due to the drift of the micro-electro-mechanical-system (MEMS) accelerometers in the low-frequency range. To solve these problems, this paper proposed dual disturbance observers (dual DOB) based on the disturbance information in the acceleration loop and the position loop. This design used two compensators to observe and compensate for the disturbances, which did not require additional sensors, and therefore did not increase system cost. Theoretical demonstrations and physical experiments showed that the designed method of the dual DOB not only improved the disturbance rejection capability of the low- and middle-frequency band of the optoelectronic stable control platform, but also improved the robustness of the system.

**Keywords:** optoelectronic stable control platform; disturbance observation and compensation; three closed-loop feedback control; disturbance rejection

---

## 1. Introduction

The tilting mirror installed on a mobile device causes visual distortion of the image, uneven image quality, poor contrast, blurred image, and sharpness under the vibration environment if a damping device is not used in the aerial imaging equipment [1]. The degradation of imaging quality leads to a decrease in the closed-loop bandwidth and the tracking accuracy of the system, which will result in the loss of tracking targets [2,3]. Therefore, improving the disturbance rejection capability of the system is key to ensure the airborne television image has a high-quality, stable tracking ability, while reducing the image blur caused by flutter [4,5]. Still, system mechanical resonance, sensor noise drift, and external vibration, caused by friction torque, wind resistance, unbalance, etc., are the main factors that limit the disturbance control ability of the stable control system. Therefore, it is of practical interest to improve the disturbance rejection ability of the stable control platform to be able to observe and compensate for the disturbance source [6].

The disturbance observer proposed by Ohnishi et al. provides a simple and effective method for obtaining disturbance, which is difficult to achieve by direct measurement [7,8]. The disturbance observer has been applied to robotics and industrial automation fields [9]. It is known from experimental research that the disturbance frequency of the optoelectronic stable control platform is usually concentrated in the low- and middle-frequency bands [10]. The three closed-loop feedback control method, which includes the acceleration loop, the velocity loop, and the position loop, is a commonly used method to improve the bandwidth and disturbance rejection capability of the system. However, this method is not designed for the suppression of the disturbance. To solve this problem, Deng Chao introduced a disturbance observer into the acceleration loop to focus on improving the disturbance rejection capability of the middle-frequency band [11]. The drift of the MEMS (micro-electro-mechanical system) accelerometers in the low-frequency range is the reason why the acceleration loop disturbance observer cannot improve the low-band disturbance rejection capability [12]. A method of adding a virtual gyroscope to suppress disturbances is proposed in the references [13], but this method needs additional sensors to measure the disturbance. At the same time, its impact on system robustness has not been proven. Li Guohui adopts a method of detecting the centroid position as a feedback signal and applies a hardware closed loop to improve the closed-loop bandwidth and disturbance rejection of the system by shortening the delay [14]. However, this method demands extraordinary equipment and high costs [15]. Advanced controllers such as LQG (linear quadratic Gaussian) [16], and others are also used to improve the closed-loop tracking performance of the system, but they all have to rely on the high-precision system dynamic models [12]. Tang Tao proposes a structure of disturbance observer based on Q filter, and the parameters of the disturbance observer were optimised by the Youla parameter method [17]. But this structure can only improve the closed-loop bandwidth and the tracking ability of the optoelectronic stable control system [18].

The CCD (charge-coupled device) sensor used in the position loop has advantages, such as high sensitivity and fast response in the low-frequency range [11]. According to those features, this paper proposed a new method, named dual disturbance observers (dual DOB), of introducing two disturbance observers in the position loop and the acceleration loop. The position loop disturbance observer can compensate for the residual disturbance that is not suppressed in the acceleration loop disturbance observer. In the position loop, this method not only removes the structural resonance of the platform but also overcomes the problem caused by the MEMS accelerometers drift in the low-frequency band. The disturbance rejection capability in the low and middle frequency is improved at the same time.

There are two main design methods of disturbance observer. One is using the inverse model and low-pass filter, the other is using model estimation and high-pass filter [19]. If observers use the nominal inverse model, they will inevitably contain differential links, which will amplify the noise, thus affecting the robustness and control accuracy of the system [19]. Therefore, in the new method of dual disturbance observers, both the position loop and the acceleration loop adopted the model estimation method to avoid the amplification of noise [20]. Theoretical demonstrations and physical experiments showed that the designed method of the dual DOB not only improved the disturbance rejection capability of the low- and middle-frequency bands of the optoelectronic stable control platform, but also improved the robustness of the system.

In this paper, the following aspects are elaborated: Section 2 models a structure of the stable control platform and offers a new system structure of dual DOB; Section 3 analyzes the stability, tracking control ability, and disturbance rejection capability of the dual DOB system; Section 4 introduces the disturbance of the acceleration loop separately; Section 5 proves the feasibility of the dual DOB system and its effectiveness in improving the disturbance rejection ability by using comparative experiments in the actual system; and Section 6 presents the conclusion of this paper.

## 2. Dual Disturbance Observers

The optoelectronic stable control platform applied to the rotating frame has the shortcomings of an extensive working range, a narrow frequency band, z slow output response, and low-tracking accuracy. Therefore, to achieve high-bandwidth and high-precision tracking control of the system under the disturbance environment, the rack and the optoelectronic stable control platform usually adopt a complex axis control. The main structure of the stable control platform is shown in Figure 1.

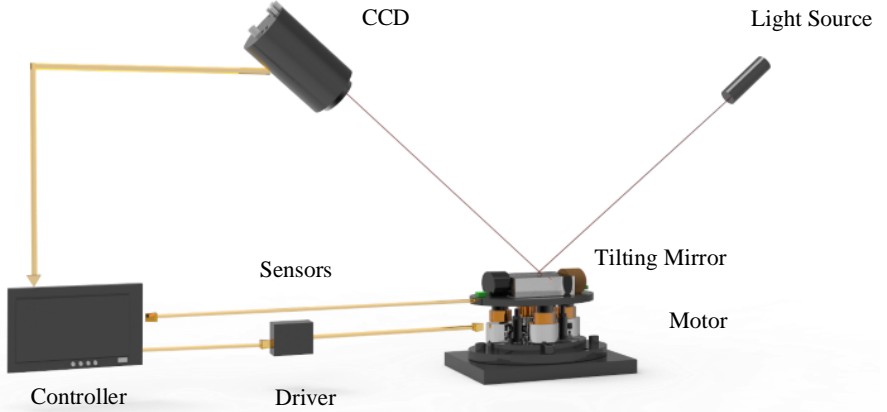

**Figure 1.** Structural sketch of stable control platform [13].

To achieve stable tracking of maneuvering targets, two main issues need to be solved [19]: one is how to ensure the stability of the optical axis and the other is target tracking technology. Stability is the precondition of tracking. Thus, better disturbance rejection capability of the platform is conducive to better tracking accuracy of the system. Based on ensuring that the platform has excellent disturbance rejection capability, the tracking performance of the system can be further improved. Due to the limitations of low sampling frequency and slow response speed of the CCD sensor for position loop, the tilt telescope cannot achieve the ideal high bandwidth. Existing optoelectronic stable control platforms mostly adopt a three closed-loop control method, especially three closed-loop feedback control methods [20] to improve the disturbance rejection ability of the system.

In Figure 2, $G_a$ is the controlled object of the acceleration loop, $C_a$ is the controller of the acceleration loop, $C_v$ is the controller of the velocity loop, and $C_p$ is the controller of the position loop.

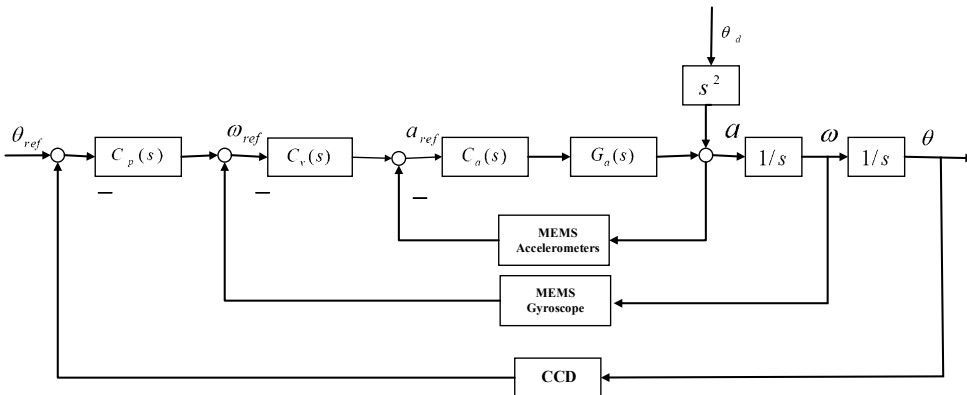

**Figure 2.** Diagram of a three closed-loop feedback control system.

According to Figure 2, it can be calculated that:

$$\theta = \frac{C_p C_v C_a G_a \frac{1}{s^2}}{1 + C_a G_a + C_v C_a G_a \frac{1}{s} + C_p C_v C_a G_a \frac{1}{s^2}} \theta_{ref} \\ + \frac{1}{1 + C_a G_a + C_v C_a G_a \frac{1}{s} + C_p C_v C_a G_a \frac{1}{s^2}} \theta_d \quad . \tag{1}$$

When the given value $\theta_{ref}$ is 0, the disturbance rejection characteristics of the three closed-loop system can be calculated as follows:

$$\begin{aligned} E &= \frac{\theta}{\theta_d} = \frac{1}{1 + C_a G_a + C_v C_a G_a \frac{1}{s} + C_p C_v C_a G_a \frac{1}{s^2}} \\ &\approx \frac{1}{1 + C_a G_a} \cdot \frac{1}{1 + \frac{1}{s} C_v} \cdot \frac{1}{1 + \frac{1}{s} C_p} \end{aligned} \quad . \tag{2}$$

It can be seen from Equation (2) that the disturbance rejection ability of the three closed-loop system is the superposition of the disturbance rejection ability of the acceleration loop, the velocity loop, and the position loop so that the disturbance rejection ability of the whole three closed-loop feedback control system depends on the design of three closed-loop controllers. The three closed-loop feedback control method can improve the disturbance rejection capability of the system. This method obtains more information about the platform itself through feedback, but it still provides an insufficient improvement in the disturbance rejection ability of the system. The visual sensor is the only device that provides target tracking information, but the information was too simple to realize our goals. To improve the tracking performance and the disturbance rejection capability of the system, it is necessary to estimate other motion information that cannot be directly measured, thereby necessitating composite feedforward control.

The perturbation torque applied to the tilting mirror on the optoelectronic stable control platform is mainly concentrated in the low-frequency and intermediate-frequency parts of 1–20 Hz [20]. To solve the problem of poor low-frequency disturbance rejection capability of the system, this paper proposed the structure of the dual disturbance observers using the acceleration loop and the position loop. The diagram of the control structure of the dual disturbance observers is shown in Figure 3. MEMS accelerometers, MEMS gyroscope, and CCD are acceleration, velocity, and position loop sensors, respectively. Their gains are determined by the scale factors of each sensor, but they were normalized in the article and are equivalent to the gain of the controller.

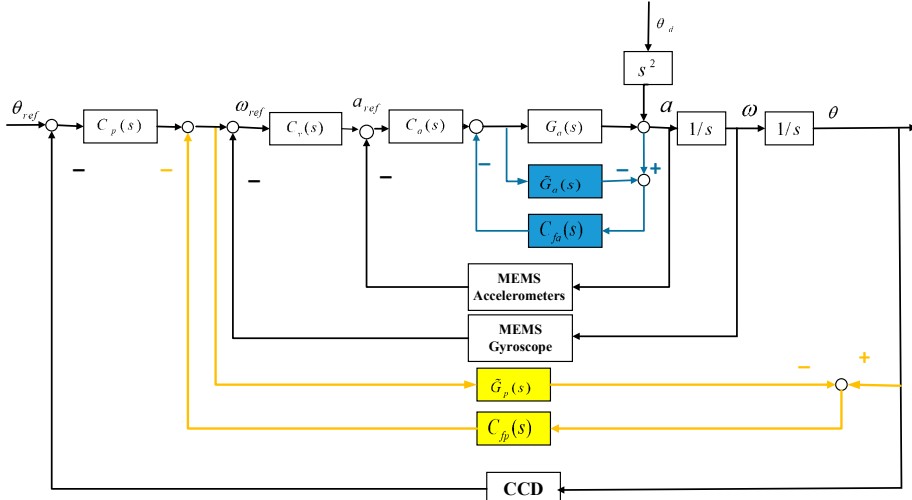

**Figure 3.** Diagram of the dual disturbance observers structure.

As shown in Figure 3, the loop indicated by the blue line is the acceleration loop disturbance compensation structure and the loop indicated by the yellow line is the position loop disturbance

compensation structure. $C_{fa}$ and $C_{fp}$ are the disturbance compensators for the acceleration loop and position loop, respectively. Their input information is the observed acceleration disturbance and position disturbance, respectively. $\widetilde{G}_a$ is the estimation model of the acceleration controlled object and $\widetilde{G}_p$ is the estimation model of the velocity controlled object.

The CCD sensor for measuring the deflection angle of the telescope applied to the position loop has a low-frequency measurement noise compared with the MEMS gyroscope of the velocity loop and does not cause integral saturation, which is why DOB was used in the position loop instead of the velocity loop. Thus, multi-information dual disturbance observers can improve the overall disturbance rejection capability by simultaneously observing the acceleration disturbance and position disturbance of the system.

The transfer function of the proposed dual disturbance observers control method can be derived as follows:

$$
\begin{aligned}
\theta = {} & \frac{C_p G_p}{1 + C_p G_p + (G_p - \widetilde{G}_p) C_{fp}} \theta_{ref} \\
& + \frac{(1 - \widetilde{G}_a C_{fa})(1 - \widetilde{G}_p C_{fp})}{(1 - \widetilde{G}_p C_{fp})\left[1 + G_a C_a + (G_a - \widetilde{G}_a) C_{fa} + G_a C_a C_v \frac{1}{s}\right] + (C_p + C_{fp}) G_a C_a C_v \frac{1}{s^2}} \theta_d,
\end{aligned}
\tag{3}
$$

where $G_P = \dfrac{C_v C_a G_a \frac{1}{s^2}}{1 + C_a G_a + (G_a - \widetilde{G}_a) C_{fa} + C_v C_a G_a \frac{1}{s}}$.

## 3. System Stability and Robustness Analysis

### 3.1. System Stability Analysis

According to the modelling of the dual DOB in Section 2, the tracking transfer function of the proposed method can be expressed as follows:

$$
T_{track} = \frac{\theta}{\theta_{ref}} = \frac{C_p G_p}{1 + C_p G_p + (G_p - \widetilde{G}_p) C_{fp}}.
\tag{4}
$$

It can be seen from Equation (4) that when the fitting result of the acceleration $\widetilde{G}_a$ is approximately equal to the actual controlled object of the acceleration $G_a$ (meaning that $\widetilde{G}_a \approx G_a$), the design of the disturbance compensator of the acceleration does not affect the stability of the system [11]. Therefore, the robustness of the system is only limited by the design parameters of the position loop disturbance compensator $C_{fp}$.

**Lemma 1.** *To ensure the stability of the optoelectronic stable control system, according to the small gain theorem, the position loop disturbance compensator should be designed to meet the following requirements:*

$$
\|C_{fp}\|_\infty < \left\| \frac{P_{initial}}{\widetilde{G}_p (1 + C_a G_a + C_v C_a G_a \frac{1}{s} - C_v C_a G_a \frac{1}{\widetilde{G}_p} \frac{1}{s^2})} \right\|_\infty,
\tag{5}
$$

*where, $P_{initial}(s) = 1 + C_a G_a + C_v C_a G_a \frac{1}{s} + C_p C_v C_a G_a \frac{1}{s^2}$.*

**Proof.** $\widetilde{G}_a$ is obtained by fitting the controlled acceleration object obtained by the system identification of the frequency response test. Therefore, we assume that $\widetilde{G}_a \approx G_a$. At this time, $G_P = \dfrac{C_v C_a G_a \frac{1}{s^2}}{1 + C_a G_a + C_v C_a G_a \frac{1}{s}}$ is the same as the controlled object of the velocity loop in the three closed loops and, thus, the acceleration loop is obtained. The introduction of the disturbance compensator does not affect the characteristics of the position loop-controlled object. □

To obtain the influence of the position loop disturbance compensator on the stability of the system, a comparison method of pole placement is adopted to compare the stable three closed-loop systems and the dual DOB system.

The pole of the three closed-loop stable control system can be expressed as:

$$P_{initial}(s) = 1 + C_a G_a + C_v C_a G_a \frac{1}{s} + C_p C_v C_a G_a \frac{1}{s^2}. \tag{6}$$

According to the modeling in Section 2, the poles of the dual DOB can be expressed as:

$$
\begin{aligned}
P_{dual} = & \quad (1 - \widetilde{G}_a C_{fa})(1 - \widetilde{G}_p C_{fp}) + G_a C_a C_v \left[ (C_p + C_{fp}) \frac{1}{s^2} + (1 - \widetilde{G}_p C_{fp}) \frac{1}{s} \right] \\
& + G_a (1 - \widetilde{G}_p C_{fp})(C_a + C_{fa}) \\
= & \quad P_{initial} \left[ 1 - \frac{\widetilde{G}_p C_{fp}(1 + C_a G_a + C_v C_a G_a \frac{1}{s} - C_v C_a G_a \frac{1}{\widetilde{G}_p} \frac{1}{s^2})}{P_{initial}} \right]
\end{aligned} \tag{7}
$$

Equation (7) is the denominator of Equation (4), which is the pole part. The result of polynomial simplification shows that the design of the acceleration loop disturbance compensator $C_{fa}$ will not affect the stability of the system, but the design of the position loop disturbance compensator $C_{fp}$ will affect the stability of the system.

Assuming that $\widetilde{s}$ is the pole of the system, the system should meet one of the following two conditions [21]:

$$P_{initial}(\widetilde{s}) = 1 + C_a G_a + C_v C_a G_a \frac{1}{s} + C_p C_v C_a G_a \frac{1}{s^2} = 0 \tag{8}$$

$$P_1(\widetilde{s}) = \left[ 1 - \frac{\widetilde{G}_p C_{fp}(1 + C_a G_a + C_v C_a G_a \frac{1}{s} - C_v C_a G_a \frac{1}{\widetilde{G}_p} \frac{1}{s^2})}{P_{initial}} \right] = 0. \tag{9}$$

When $C_{fp} = 0$, then $P_{dual} = P_{initial}$. When the design of the three closed-loop system keeps the system stable, the value of Equation (9) must be such that the portion remains stable.

If $C_{fp} \neq 0$, the change in $C_{fp}$ will cause $P_{dual}$ to produce a root trajectory. According to system stability theory, the system can only be stable when the root trajectory does not pass through the imaginary axis and the system can be stabilized when the system does not have the root on the right side of the imaginary axis. Therefore, the system can be stabilized only when the root trajectory $P_{dual}$ is entirely on the left side of the imaginary axis. Therefore, $P_1(s)$ should meet the conditions shown as Equation (10):

$$\left\| 1 - \frac{\widetilde{G}_p C_{fp}(1 + C_a G_a + C_v C_a G_a \frac{1}{s} - C_v C_a G_a \frac{1}{\widetilde{G}_p} \frac{1}{s^2})}{P_{initial}} \right\|_\infty \neq 0. \tag{10}$$

That is to say:

$$\left\| \frac{\widetilde{G}_p C_{fp}(1 + C_a G_a + C_v C_a G_a \frac{1}{s} - C_v C_a G_a \frac{1}{\widetilde{G}_p} \frac{1}{s^2})}{P_{initial}} \right\|_\infty \neq 1. \tag{11}$$

According to the frequency domain transformation relationship $s = j\omega$, Equation (11) should be independent of any frequency, so we divided Equation (11) into the following two cases:

First case:

$$\left\| \frac{\widetilde{G}_p C_{fp}(1 + C_a G_a + C_v C_a G_a \frac{1}{s} - C_v C_a G_a \frac{1}{\widetilde{G}_p} \frac{1}{s^2})}{P_{initial}} \right\|_\infty > 1. \tag{12}$$

Since the bandwidth of the system is limited when the angular rate $\omega$ approaches infinity, the establishment of Equation (12) cannot be guaranteed, so this situation cannot be achieved.

Second case:

$$\left\| \frac{\widetilde{G}_p C_{fp}(1 + C_a G_a + C_v C_a G_a \frac{1}{s} - C_v C_a G_a \frac{1}{\widetilde{G}_p} \frac{1}{s^2})}{P_{initial}} \right\|_\infty < 1. \tag{13}$$

Simplified:

$$\|C_{fp}\|_\infty < \left\| \frac{P_{initial}}{\widetilde{G}_p(1 + C_a G_a + C_v C_a G_a \frac{1}{s} - C_v C_a G_a \frac{1}{\widetilde{G}_p} \frac{1}{s^2})} \right\|_\infty . \tag{14}$$

When the disturbance compensator of the position loop satisfies Equation (14), the tracking performance of the system incorporating the dual DOB can be described as follows:

$$T_{track} = \frac{\theta}{\theta_{ref}} \approx \frac{C_p C_v C_a G_a \frac{1}{s^2}}{1 + C_a G_a + C_v C_a G_a \frac{1}{s} + C_p C_v C_a G_a \frac{1}{s^2}}. \tag{15}$$

Comparing Equations (15) and Equations (1), it can be seen that the position loop disturbance compensator that satisfies the condition $\widetilde{G}_a \approx G_a$ does not affect the tracking performance of the stable control platform.

### 3.2. System Robustness Analysis

Robustness is an important factor affecting system tracking performance. In order to calculate the role of dual DOB in improving the system's internal model of anti-interference ability, the uncertainty of the model was defined as $\Delta G_a$. When the model of the platform changes, the accelerometer object can be represented as $G_a + \Delta G_a$. The change in the system output caused by the change in the controlled object can be expressed by the sensitivity function. The sensitivity function can describe the change in the system output caused by the change in the controlled object and it reflects the robustness of the system [22]. When the model has a small uncertainty, the sensitivity of the pure three closed-loop system and the dual disturbance observer system can be expressed as:

$$
\begin{aligned}
S_{initial} &= \frac{[D'_1(s) - D_1(s)]/D_1(s)}{\Delta G_a(s)/G_a(s)} \\
&= \frac{1}{1 + C_a(G_a + \Delta G_a) + C_v C_a(G_a + \Delta G_a)\frac{1}{s} + C_p C_v C_a(G_a + \Delta G_a)\frac{1}{s^2}} \\
&\approx \frac{1}{1 + C_a G_a + C_v C_a G_a \frac{1}{s} + C_p C_v C_a G_a \frac{1}{s^2}}
\end{aligned}
\tag{16}
$$

$$
\begin{aligned}
S_{dualDOB} &= \frac{[D'_2(s) - D_2(s)]/D_2(s)}{\Delta G_a(s)/G_a(s)} \\
&= \frac{(1 - \widetilde{G}_a C_{fa})(1 - \widetilde{G}_p C_{fp})}{(1 - \widetilde{G}_p C_{fp})[1 + (G_a + \Delta G_a)C_a + ((G_a + \Delta G_a) - \widetilde{G}_a)C_{fa} + (G_a + \Delta G_a)C_a C_v \frac{1}{s}] + (C_p + C_{fp})(G_a + \Delta G_a)C_a C_v \frac{1}{s^2}} \\
&\approx \frac{(1 - \widetilde{G}_a C_{fa})(1 - \widetilde{G}_p C_{fp})}{(1 - \widetilde{G}_p C_{fp})[1 + G_a C_a + (G_a - \widetilde{G}_a)C_{fa} + G_a C_a C_v \frac{1}{s}] + (C_p + C_{fp})G_a C_a C_v \frac{1}{s^2}} \\
&\approx \frac{(1 - \widetilde{G}_a C_{fa})}{(1 + G_a C_a + C_{fa} + G_a C_a C_v \frac{1}{s}) + \frac{(C_p + C_{fp})}{(1 - \widetilde{G}_a C_{fa})} G_a C_a C_v \frac{1}{s^2}} \\
&= \frac{1}{(1 + G_a C_a + C_{fa} + G_a C_a C_v \frac{1}{s}) + \frac{(C_p + C_{fp})}{(1 - \widetilde{G}_a C_{fa})} G_a C_a C_v \frac{1}{s^2}} - \frac{\widetilde{G}_a C_{fa}}{(1 + G_a C_a + C_{fa} + G_a C_a C_v \frac{1}{s}) + \frac{(C_p + C_{fp})}{(1 - \widetilde{G}_a C_{fa})} G_a C_a C_v \frac{1}{s^2}}
\end{aligned}
\tag{17}
$$

where,

$$D'_1(s) = \frac{C_p C_v C_a(G_a + \Delta G_a)\frac{1}{s^2}}{1 + C_a(G_a + \Delta G_a) + C_v C_a(G_a + \Delta G_a)\frac{1}{s} + C_p C_v C_a(G_a + \Delta G_a)\frac{1}{s^2}}, \tag{18}$$

$$D_1(s) = \frac{C_p C_v C_a G_a \frac{1}{s^2}}{1 + C_a G_a + C_v C_a G_a \frac{1}{s} + C_p C_v C_a G_a \frac{1}{s^2}}, \tag{19}$$

$$D'_2(s) = \frac{C_p C_v C_a (G_a + \Delta G_a) \frac{1}{s^2}}{(1 - \widetilde{G}_a C_{fa})(1 - \widetilde{G}_p C_{fp}) + (G_a + \Delta G_a) C_a C_v \left[ (C_p + C_{fp}) \frac{1}{s^2} + (1 - \widetilde{G}_p C_{fp}) \frac{1}{s} \right] + (G_a + \Delta G_a)(1 - \widetilde{G}_p C_{fp})(C_a + C_{fa})}, \tag{20}$$

$$D_2(s) = \frac{C_p C_v C_a G_a \frac{1}{s^2}}{(1 - \widetilde{G}_a C_{fa})(1 - \widetilde{G}_p C_{fp}) + G_a C_a C_v \left[ (C_p + C_{fp}) \frac{1}{s^2} + (1 - \widetilde{G}_p C_{fp}) \frac{1}{s} \right] + G_a (1 - \widetilde{G}_p C_{fp})(C_a + C_{fa})}, \tag{21}$$

where Equation (16) is the robust sensitivity function of the three closed-loop system and Equation (17) is the sensitivity function of the dual DOB system.

From the comparison with the sensitivity functions, it is clear that conclusions can be drawn that $|S_{dualDOB}| < |S_{initial}|$. Therefore, the introduction of a dual disturbance observer can improve the robust stability of the system and reduce the influence of the model perturbation.

## 4. Design of Dual Disturbance Compensators

In the previous section, we discussed the impact of the introduction of a dual disturbance observer on system stability and robustness. In this section, the specific design method of the acceleration loop and the position loop disturbance observer is introduced.

According to Equation (3), we can get the disturbance rejection function of the dual disturbance observer, as shown in Equation (22):

$$\begin{aligned} E_{dualDOB} &= \frac{(1 - \widetilde{G}_a C_{fa})(1 - \widetilde{G}_p C_{fp})}{(1 - \widetilde{G}_p C_{fp}) \left[ 1 + G_a C_a + (G_a - \widetilde{G}_a) C_{fa} + G_a C_a C_v \frac{1}{s} \right] + (C_p + C_{fp}) G_a C_a C_v \frac{1}{s^2}} \\ &\approx \frac{(1 - \widetilde{G}_a C_{fa})(1 - \widetilde{G}_p C_{fp})}{1 + C_p G_p} \end{aligned} \tag{22}$$

It can be seen from the Equation (22) that the improvement in the disturbance rejection capability of the dual disturbance observer mechanism is mainly related to the design of the disturbance compensator $C_{fa}$ and $C_{fp}$. Therefore, we discussed the disturbance rejection capability of the acceleration ring's and the position loop's disturbance compensator separately. Suppose that the disturbance rejection functions of the acceleration loop and the position loop, respectively, are:

$$\hat{E}_a = 1 - \widetilde{G}_a C_{fa} \text{ and} \tag{23}$$

$$\hat{E}_p = 1 - \widetilde{G}_p C_{fp}. \tag{24}$$

### 4.1. Acceleration Loop Disturbance Compensator

According to the analysis in Section 3, when the fitting model of the acceleration loop is approximately equal to the acceleration ring object, it means that $\widetilde{G}_a \approx G_a$. The introduction of the acceleration disturbance observer does not affect the stability and closed-loop tracking capability of the system. The role of the disturbance compensator is to take the observed disturbance as the input and feed the compensated output back into the velocity loop. After observing the disturbance, the critical issue is the design of the disturbance compensator.

Therefore, for the design of the acceleration loop disturbance compensator, the idea was adopted that, when $C_{fa}$ satisfies Equation (25), the system can achieve the theoretical optimal disturbance rejection:

$$1 - \widetilde{G}_a C_{fa1} = 0 \rightarrow C_{fa1} = \frac{1}{\widetilde{G}_a} = \frac{\left( \frac{s^2}{\omega_n^2} + \frac{2\xi}{\omega_n} s + 1 \right)(T_a s + 1)}{K_a s^2}, \tag{25}$$

where $\widetilde{G}_a = \frac{K_a s^2}{\left( \frac{s^2}{\omega_n^2} + \frac{2\xi}{\omega_n} s + 1 \right)(T_a s + 1)}$, $K_a$ is the gain in the acceleration controlled object, $\omega_n$ is the turning frequency of the second-order link of the denominator of the acceleration controlled object, $\xi$ is the parameter in the second-order link of the denominator of the acceleration controlled object, and $T_a$ is

the parameter in the first-order lag link of the denominator of the acceleration controlled object. The above four parameters can be identified for the purpose of the controlled acceleration object.

The Bode diagram of the ideal acceleration loop disturbance compensator $C_{fa1}$ in the first case is shown in Figure 4.

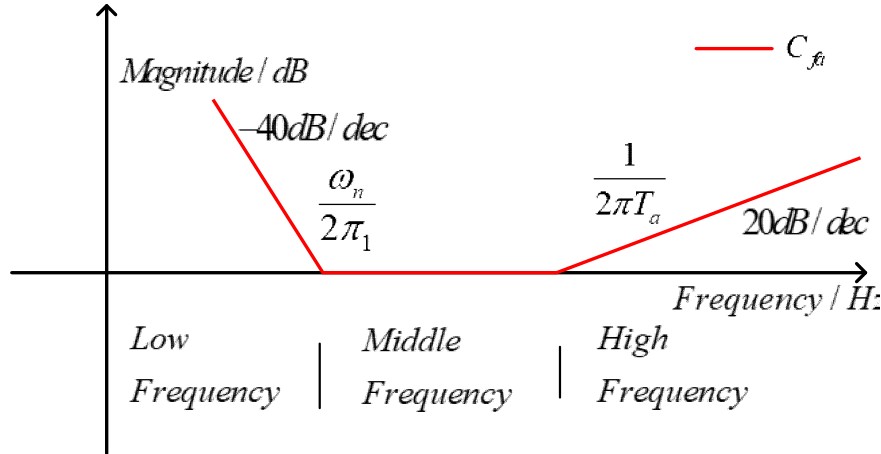

**Figure 4.** Bode diagram of the ideal acceleration-loop disturbance compensator.

It can be seen from Figure 4 that the second integration of the disturbance observer designed according to Equation (25) will amplify the measurement noise of the accelerometer in the low-frequency band. In this design, the low-frequency noise of the accelerometer is too large at the low frequency, which will cause the low-frequency integral saturation of the compensator. At the same time, the passive interference suppression of the optoelectronic stable control platform itself will also suppress the disturbance of the compensatory effect. Therefore, in order to ensure the stability of the optoelectronic stable control platform and a better suppression of the mid-range disturbance, the disturbance compensator of the acceleration loop was designed in the form of a lead-lag controller. In the second case, the acceleration disturbance compensator was designed according to the constraint of $\left| 1 - \widetilde{G}_a C_{fa} \right| < 1$:

$$C_{fa2} = \frac{K_{fa}(T_a s + 1)}{s^2 + 2\xi_{fa}\omega_{fa} + \omega_{fa}^2}. \tag{26}$$

In Equation (26), where $(T_a s + 1)$ is used to compensate phase lag, the second-order resonance element $(s^2 + 2\xi_{fa}\omega_{fa} + \omega_{fa}^2)$ is used to filter the high-frequency noise of accelerometers and partly compensate for the quadratic integration.

The Bode diagram of the improved acceleration loop disturbance compensator $C_{fa2}$ in the second case is shown in Figure 5.

Since the low-frequency sensitivity of the low-frequency accelerometer is low, the compensation at low frequencies can be ignored. The phase lag used to compensate for the controlled acceleration object $(T_a s + 1)$ is located in the second-order link of the denominator to filter out high-frequency noise.

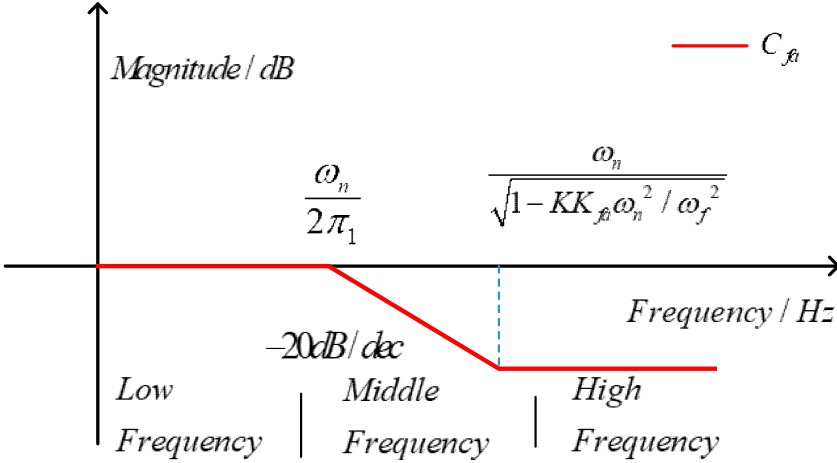

**Figure 5.** Bode diagram of original lead-lag disturbance compensator.

The acceleration loop disturbance rejection function under this design method is in the form of a trap. The designer adjusts the frequency of the notch and the notch width through design parameters:

$$\hat{E}_a = 1 - \widetilde{G}_a C_{fa} = \frac{(\frac{1}{\omega_n^2} - \frac{K_a K_{fa}}{\omega_{fa}^2})s^2 + 2(\frac{\xi_a}{\omega_n} + \frac{\xi_{fa}}{\omega_{fa}^2})s + 1}{\frac{s^2}{\omega_n^2} + 2(\frac{\xi_a}{\omega_n} + \frac{\xi_{fa}}{\omega_{fa}^2})s + 1}. \tag{27}$$

The acceleration loop disturbance rejection function $E_a$ is similar in form to the groove and its Bode diagram is shown in Figure 6.

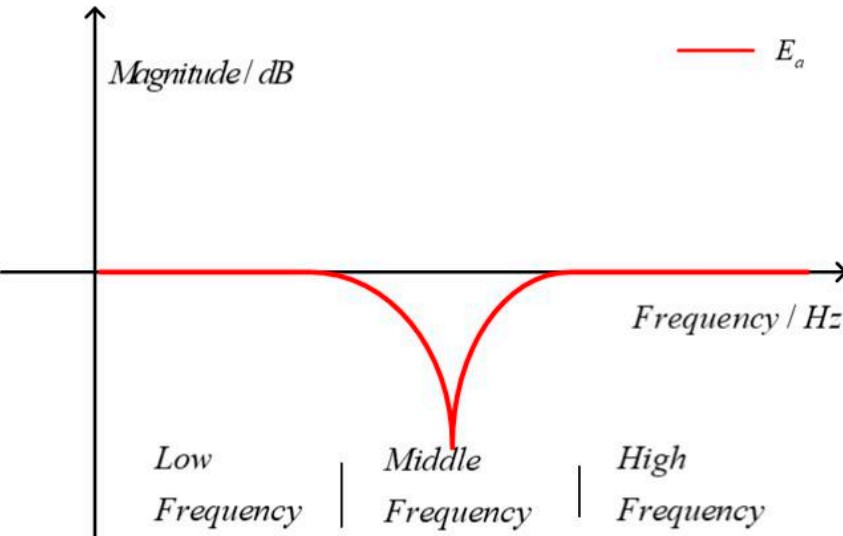

**Figure 6.** Bode diagram of disturbance rejection function of acceleration ring.

It can be seen from the Bode diagram of the acceleration ring disturbance rejection function that the main function of the disturbance compensation of the acceleration loop is to suppress the disturbance of the middle-frequency band. To improve the disturbance rejection capability of the low-frequency band simultaneously, a disturbance compensator for the position loop is introduced.

*4.2. Position Loop Disturbance Compensator*

Due to the existence of MEMS low-frequency drift, a simple acceleration loop DOB cannot improve the system's low-frequency disturbance suppression capability. Therefore, adding a disturbance

observer and compensator in the position loop is adopted to suppress the low frequency disturbance of the system. The first design method is the same as the first design of the acceleration loop disturbance compensator. When $C_{fp}$ satisfies Equation (28), the disturbance rejection function of the dual disturbance observer is theoretically optimal, that is, it approaches zero:

$$1 - \widetilde{G}_p C_{fp1} = 0 \rightarrow C_{fp1} = \frac{1}{\widetilde{G}_p}. \tag{28}$$

According to the stability analysis of the system in Section 3, the disturbance compensator of the position loop should satisfy the conditions in Equation (14). To meet the stability requirements of the system, the disturbance compensator of the position loop should be guaranteed to suppress the low-frequency participation disturbance. For this role, we considered the second design method: design a position loop disturbance rejection function as the form of the lead-lag corrector.

In the low and mid bands, the estimate of the controlled object of the position loop $\widetilde{G}_p$ can be described as Equation (29):

$$\widetilde{G}_p = \frac{1}{s}. \tag{29}$$

The disturbance rejection characteristic of the position loop can be simplified to Equation (30):

$$\hat{E}_p = 1 - \widetilde{G}_p C_{fp2} = \frac{K_p(T_1 s + 1)}{(T_2 s + 1)}, \tag{30}$$

where $T_1 > T_2$.

The transfer function Bode diagram of the position loop disturbance compensator is shown in Figure 7.

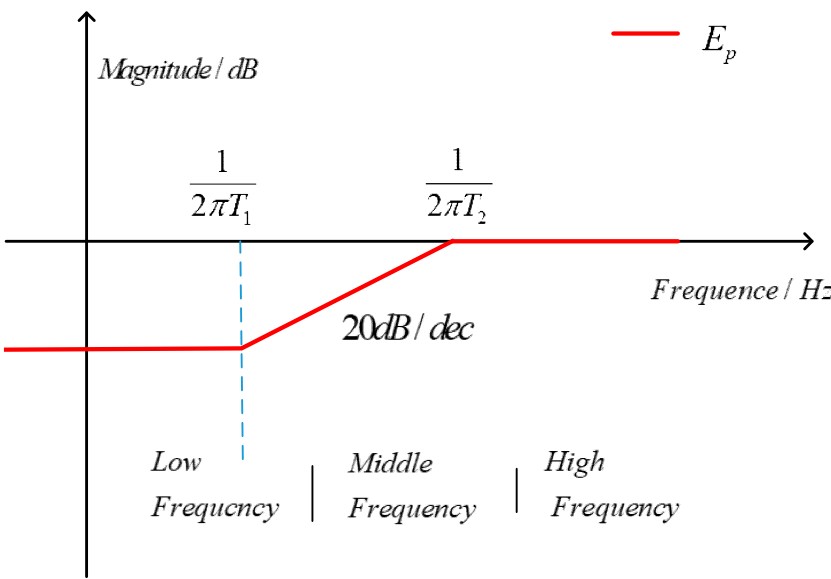

**Figure 7.** Bode diagram of the disturbance rejection function of the position loop.

The position loop disturbance compensator is now in the form of the Equation (31):

$$C_{fp2} = \frac{(1 - \hat{E}_p)}{\widetilde{G}_p}. \tag{31}$$

It can be seen from the disturbance rejection function of the position loop that the addition of the position loop disturbance compensator mainly improves the performance of the low frequency

and the intermediate frequency and, at the same time, compensates for the low-frequency disturbance rejection capability that the acceleration loop fails to improve.

In summary, we designed the disturbance compensator for the acceleration loop and the position loop, respectively, as shown in Equations (32) and (33):

$$C_{fa} = \frac{K_{fa}(T_a s + 1)}{s^2 + 2\xi_{fa}\omega_{fa} + \omega_{fa}{}^2} \text{ and} \tag{32}$$

$$C_{fp} = \frac{(1 - \hat{E}_p)}{\widetilde{G}_p} = \frac{(1 - \frac{K_p(T_1 s + 1)}{(T_2 s + 1)})}{\widetilde{G}_p} = \frac{\left[(T_2 s + 1) - K_p(T_1 s + 1)\right]}{\widetilde{G}_p(T_2 s + 1)}. \tag{33}$$

## 5. Experiments

To verify the improvement of the disturbance rejection performance of the dual DOB on the stable control platform, we used the experimental devices shown in Figure 8 for verification.

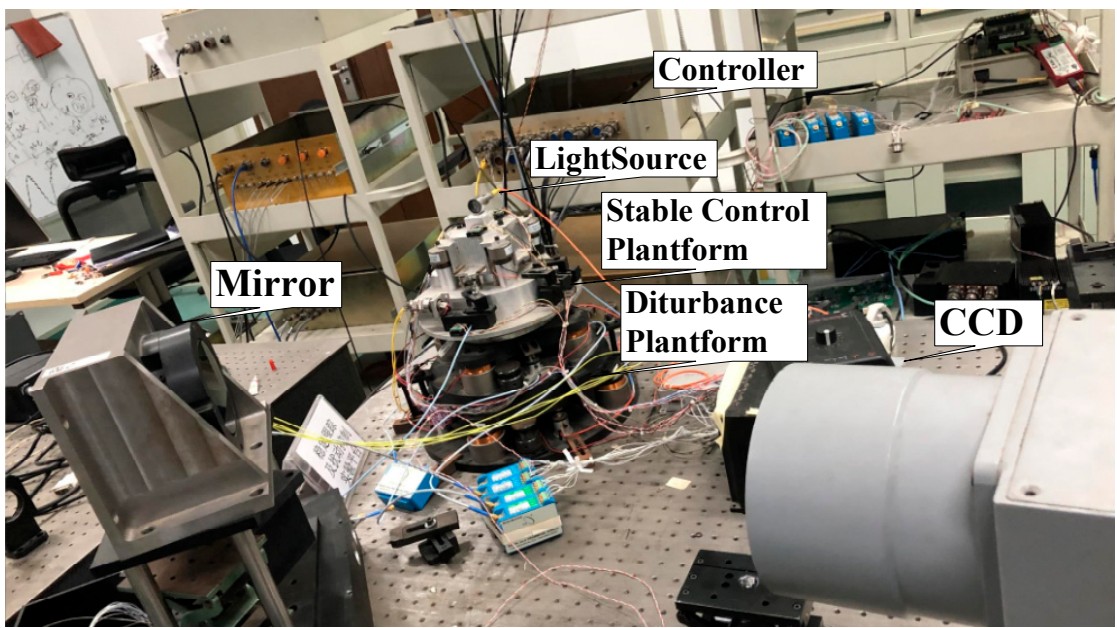

**Figure 8.** Experimental Platform.

Two motion platforms superimposed the experimental system. The lower platform was used to simulate the platform disturbance while the upper platform was used to stabilize the laser light. The inertia-stabilized platform was equipped with MEMS gyroscopes and MEMS accelerometers. Inside the platform, the MEMS accelerometer was the feedback component of the acceleration loop and the MEMS gyroscope was the feedback component of the velocity loop. The CCD acted as a position loop sensor to detect the position signal of the beam. The sampling frequency of the CCD sensor was 100 Hz, with a hysteresis of two frames, while the sampling frequencies of the MEMS accelerometer and the MEMS gyroscope were both 5000 Hz.

When the closed-loop velocity sweep of the stable platform was performed, the disturbance platform was locked. The stability test was to drive the signal to the disturbance platform in the case of a stable platform closed loop and compare the gyroscope output signal of the stable platform with the sign of the disturbance platform speedometer.

By sweeping and fitting the acceleration object in the 1–1000 Hz range, the result shown in Figure 9 was obtained.

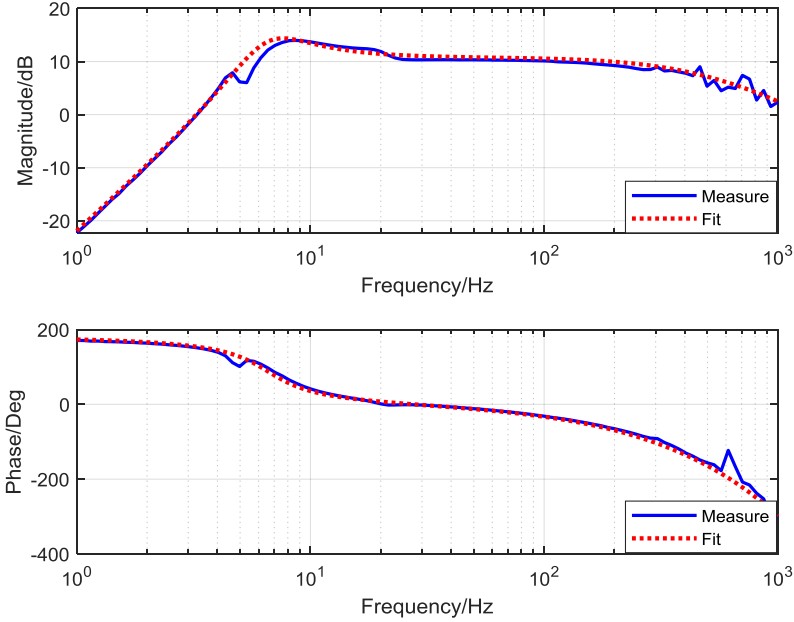

**Figure 9.** The open-loop characteristic acceleration response of the controlled object.

The transfer function of the controlled object obtained by the system identification was as shown in Equation (34):

$$\widetilde{G}_a = \frac{3.439s^2}{0.00035s^3 + 1.01s^2 + 29.63s + 1720}. \tag{34}$$

Since the acceleration closed-loop transfer function of the stable inertial platform appeared as a second-order oscillation link, there is a significant phase lag near the resonance peak of the transfer function, which will seriously affect the closed-loop bandwidth of the velocity loop. The design of the acceleration loop controller is shown in Equation (35):

$$C_a = \frac{107(0.0003s + 1)}{(0.67s + 1)(0.00019s + 1)}. \tag{35}$$

The controller of the velocity loop and position loop used the traditional (proportion-integral) PI or (proportion-integral-derivative) PID controller form, and the expressions are shown in Equations (36) and (37):

$$C_v = \frac{0.5(0.2s + 1)}{s + 1} \text{ and} \tag{36}$$

$$C_p = \frac{5(3.8s + 1)}{s + 1}. \tag{37}$$

The design of the disturbance compensator of the acceleration loop was the same as before. The design of the position loop disturbance compensator also needed to estimate the model of the position-controlled object. By sweeping and fitting the position object in the 1 Hz–100 Hz frequency band, the results were obtained as shown as Figure 10.

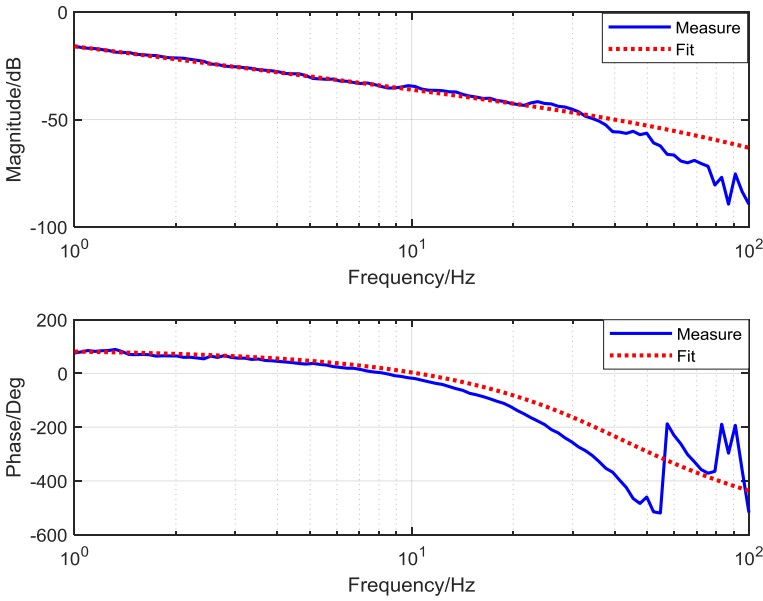

**Figure 10.** The open-loop characteristic position response of the controlled object.

The transfer function of the velocity controlled object obtained by the system identification is as shown in Equation (38):

$$\widetilde{G}_p = \frac{1}{s(0.0006s + 1)(0.003s + 1)}. \tag{38}$$

According to the design method of the dual DOB in Section 4 and the controlled object measured by the frequency sweep test, the disturbance compensator of the acceleration loop and the position loop was designed. The parameter design is as shown in Equations (39) and (40):

$$C_{fa} = \frac{6.3\,s + 18000}{s^2 + 600s + 63165} \text{ and} \tag{39}$$

$$C_{fp} = \frac{0.6s}{0.0193s + 1}. \tag{40}$$

The time-domain error comparison of the three structures with the disturbance frequency of 1, 2, 3, and 10 Hz is shown in Figure 11.

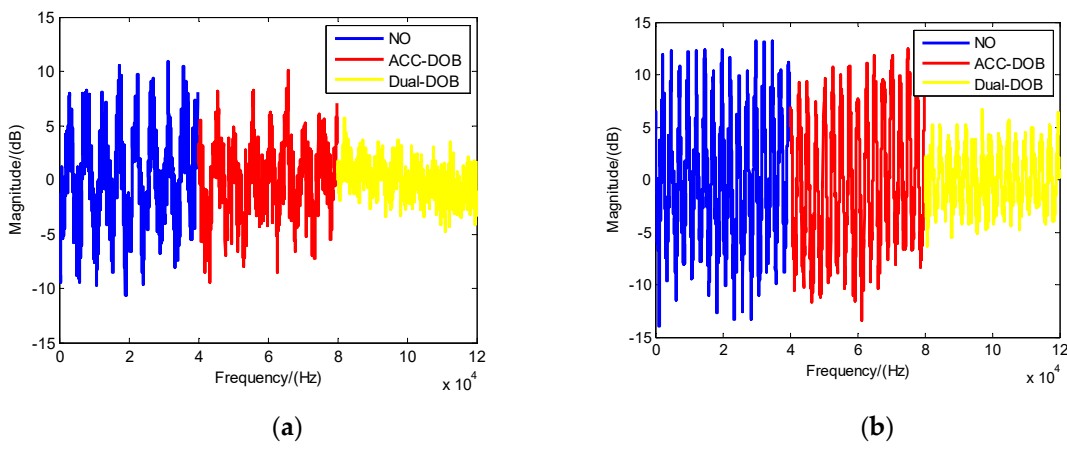

(**a**)                                     (**b**)

**Figure 11.** *Cont.*

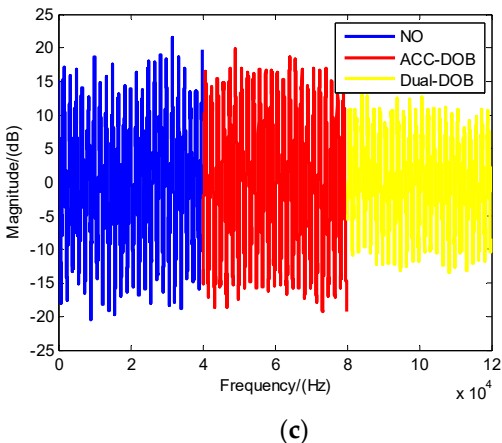
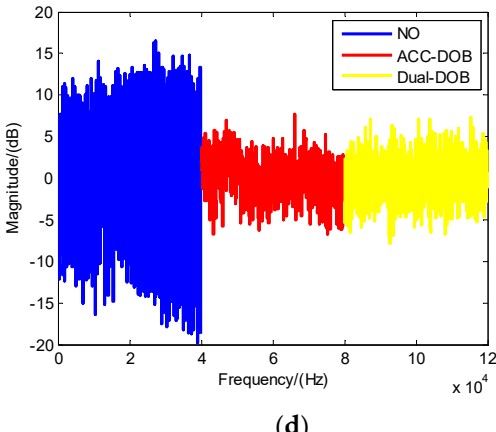

**Figure 11.** Time-domain error comparison of three structures. (**a**) Time-domain error comparison at a disturbance frequency of 1 Hz; (**b**) time-domain error comparison at a disturbance frequency of 2 Hz; (**c**) time-domain error comparison at a disturbance frequency of 3 Hz; and (**d**) time-domain error comparison at a disturbance frequency of 10 Hz.

The results of the perturbation experiments with four given frequencies from 1, 2, 3, and 10 Hz are shown above. According to the RMS (root mean square) and peak data of the three methods of system disturbance error at four different frequencies in Table 1, the dual-loop DOB had a significant reduction in the mean square value and peak value at 1, 2, and 3 Hz versus NO DOB and only the acceleration loop disturbance observer. The RMS and peak value at 10 Hz had slight disadvantages compared with ACC DOB, but the effect was not much different and the error suppression effect was also significantly better than that without DOB.

**Table 1.** Time-domain error, mean square, and peak comparison of three structures with disturbance frequencies of 1, 2, 3, and 10 Hz.

| Disturbance Frequency | RMS Contrast | | | Max Peak Contrast | | |
|---|---|---|---|---|---|---|
| | **NO-DOB** | **ACC-DOB** | **Dual-DOB** | **NO-DOB** | **ACC-DOB** | **Dual-DOB** |
| 1 Hz | 4.2238 | 3.2079 | 1.5420 | 10.9626 | 10.1216 | 5.6873 |
| 2 Hz | 5.8134 | 6.1991 | 2.3941 | 13.2282 | 12.4643 | 6.6835 |
| 3 Hz | 8.8128 | 11.1117 | 6.7706 | 21.6176 | 19.8468 | 12.9212 |
| 10 Hz | 8.7627 | 2.7128 | 2.7698 | 16.4549 | 7.7296 | 7.2940 |

The designed dual DOB was used to verify the disturbance rejection effect on the experimental platform. The obtained system disturbance rejection characteristics are shown in Figure 12.

Based on the above experimental results, it can be seen that the addition of the dual DOB significantly improved the disturbance rejection capability from 1 to 15 Hz compared with the pure three-closed system. In contrast with the results of the single comparison loop (single acceleration loop) disturbance observer, the dual DOB not only retained significant improvement in the disturbance rejection ability of the acceleration disturbance observer at around 10 Hz, but also of the system in the frequency range of 1 to 10 Hz. The disturbance rejection capability improved from −5 to −10 dB. Therefore, the experimental results show that the results of the dual DOB have a significant effect on the improvement of the system's disturbance rejection ability.

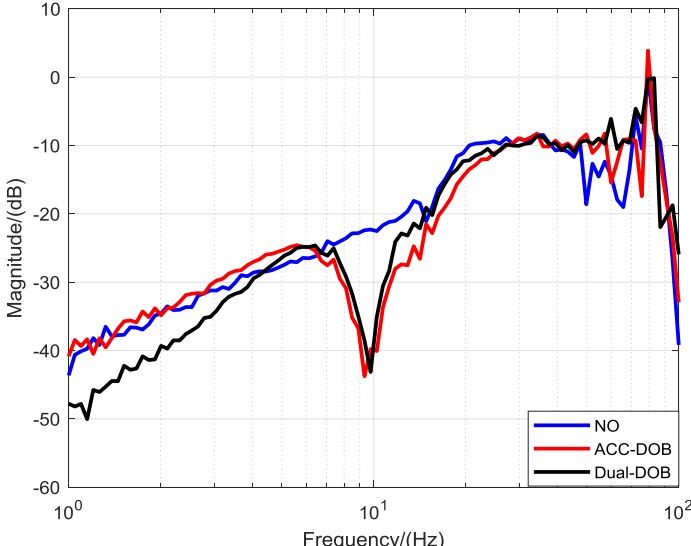

**Figure 12.** Comparison of the disturbance rejection characteristics of three structures.

## 6. Conclusions

To improve the disturbance rejection capability of the optoelectronic stable control platform in aeronautical imaging equipment, this paper proposed dual disturbance observers based on acceleration loop and position loop disturbance information. Compared with the traditional three closed-loop feedback control systems, this structure significantly improves the low-frequency and medium-frequency disturbance rejection capability. Compared with the single-acceleration loop disturbance observer, the position loop disturbance compensator effectively improves the disturbance rejection capability of the low frequency by suppressing residual disturbance.

Also, the influence of the introduction of the dual DOB on the tracking performance and system stability of the system was also analyzed. The design range of the disturbance compensator was given and the robustness of the structure of the system was proved to be enhanced. At the same time, the experimental results of the disturbance rejection characteristics of the whole frequency range and the mean square error data at different frequencies showed that the dual DOB structure effectively improves the disturbance rejection capability of the system within the designed frequency.

**Author Contributions:** Conceptualization, H.Z. and Y.M.; Data curation, H.Z. and J.D.; Formal analysis, H.Z.; Investigation, H.Z.; Methodology, H.Z. and H.L.; Project administration, Y.M.; Resources, Y.M.; Software, H.Z. and J.D.; Validation, Y.M. and H.L.; Writing—original draft, H.Z.; Writing—review & editing, H.Z. and H.L. All authors have read and agreed to the published version of the manuscript.

**Funding:** This research received no external funding.

**Conflicts of Interest:** The authors declared no conflicts of interest.

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
