# Peer review of "Three Closed-Loop Feedback Control System with Dual Disturbance Observers of an Optoelectronic Stable Control Platform"

_electronics, doi:10.3390/electronics9020359_

Round 1
Reviewer 1 Report
380 Conflicts of Interest: The authors declared no conflflict of interest.
To be changed "conflflict" into "conflict".
Author Response
Thank you for your careful review and detailed comments.Please see the attachment for detailed response.

Reviewer 2 Report
The authors in the summary and introduction point out that the main problem is MEMS drift. This phenomenon is no longer mentioned later in the article. There are no conclusions.
What results are quoted on line 162?
How the uncertainty of the model was determined (line 194)?
The authors mention that: “After observing the disturbance, the critical issue is the design of the disturbance compensator” (line 223). Has a specific quality indicator been used?
line 237-238 - This sentence is not clear.
The system described in the article is an extension of the system described in reference 19. Figures 1 and 2 are the same. In the reviewer's opinion, the results of the experiment should also include a comparison to the results from reference 19. It is not enough to compare different methods with each other. Comparisons with the results of other articles should be provided.
The text requires errors correction, e.g.:
line 39 – „sourcen”
line 53 – „applys”
line 92 – „manoeuvring”
line 131 – „diaturbance”
Author Response

(The authors gave the same response as above.)

Reviewer 3 Report
Comments to the authors:
1. The authors shall justify the significance of this paper. Frequency-domain control and observer design is not the main-stream selection compared with state-space methods due to poor accuracy and robustness. The authors shall compare with existing results in the similar topics using state-space models, and explain why their method is more advanced.
2. The control design must be clearly defined. For example:
In Fig. 2, what are the gains for "MEMS Accelerometers", "MEMS Gyroscope" and "CCD"? It looks like all the gains have been set as 1 according to Equation (1), but there is no clear clue from the paper. The authors shall provide more clear and detailed proof to justify their design as shown in Fig. 3. The authors shall provide detailed design instructions for the parameters on Line 244, Page 10.3. Some minor format / language / typo issues need to be addressed. For example,
Line 35, Page 1, there shall be a space between "flutter" and "[4, 5]. Line 229 on Page 10, the Greek letter "\zeta" shall be "\xi"? In the reference section, the citation format must be unified.
Author Response

(The authors gave the same response as above.)

Round 2
Reviewer 3 Report
The revised version has shown significant improvement.
Author Response
Thank you very much for your careful review. We further examined the article and modified the grammar and presentation issues in the article. At the same time, the supplementary explanation of the design structure is provided in Lines 134-139. We will conduct further research on your question in the follow-up work, and thank you again for your valuable comments.
